# Effects of Various Numbers of Runs on the Success of Hamstring Injury Prevention Program in Sprinters

**DOI:** 10.3390/ijerph19159375

**Published:** 2022-07-30

**Authors:** Yusaku Sugiura, Kazuhiko Sakuma, Shimpei Fujita, Kazuhiro Aoki, Yuji Takazawa

**Affiliations:** 1Sport Science, Meikai University, Urayasu 279-8550, Japan; 2Graduate School of Health and Sports Science, Juntendo University, Inzai 270-1695, Japan; kzsakuma@juntendo.ac.jp (K.S.); k-aoki@juntendo.ac.jp (K.A.); 3College of Health and Welfare, J.F. Oberlin University, Machida 194-0294, Japan; fujita_s@obirin.ac.jp; 4Department of Sports Medicine, Graduate School of Medicine, Health and Sports Science, Juntendo University, Tokyo 113-0033, Japan; ytakaza@juntendo.ac.jp

**Keywords:** hamstring injury, muscle fatigue, sprinters, maximal running, supramaximal running

## Abstract

Studies have not adequately addressed the influence of fatigue, which is considered a major risk factor for hamstring injuries. Therefore, this study aimed to clarify how a muscle fatigue condition affects the success of hamstring injury prevention programs in sprinters. The study subjects were 613 collegiate male sprinters. They employed submaximal/maximal running for a large number of runs and supramaximal running for a small number of runs in daily training. The hamstring injury prevention program had become the most effective strategy in the past 24 seasons of track and field for preventing hamstring injuries. The number of sprinters who experienced hamstring injuries in three periods over the 24 seasons was recorded. The incidents of hamstring injuries during supramaximal running per athlete-seasons were 137.9, 60.6, and 6.7 for Periods I, II, and III, respectively, showing a significant decline (*p* < 0.01). Furthermore, the incidents of hamstring injuries during submaximal and maximal running per season showed no significant change. The results of this study indicate that by inducing muscle fatigue, a small number of runs makes hamstring injury prevention programs effective.

## 1. Introduction

Most researchers on hamstring injury prevention focus on ball sports (e.g., soccer and football) [1,2,3,4,5] and less on sprinters [6,7]. These studies show the effectiveness of a combination of eccentric hamstring strength training and intervention to prevent hamstring injury. According to Sugiura et al. [6,7], the number of hamstring injuries in sprinters decreased when agility, eccentric strength, and flexibility trainings were incorporated into the existing prevention concentric strength training programs.

When the number of risk factors exceeds the threshold, it leads to injury, and some factors are more predictive than others [8]. In a theoretical model, Worrell [9] suggested that a combination of abnormalities associated with strength, flexibility, and warm-up methods and fatigue increases the risk of a hamstring injury. Agre [10] further lists some possible causes associated with the development of hamstring injury when running or sprinting, such as inadequate flexibility of the muscles, inadequate muscle strength, and dys-synergic muscle contraction.

During competition, hamstring injuries tend to occur at the end of both halves of the match [11,12]. Wood et al. [12] reported that approximately half (47%) of hamstring injuries sustained during matches occurred in the last third of the first and second halves of the match, probably because of fatigue-related consequences. The role of muscle fatigue has previously been discussed and identified as a factor in causing injury [13,14].

During practice sessions [15], the most commonly observed incidences of hamstring injury were from men’s indoor track (5.93), followed by soccer (5.01), outdoor track (3.91), and football (3.82). When adding the index of indoor and outdoor tracks, sprinters show almost the highest incidences of hamstring injuries. The majority of hamstring injuries in sprinters occur when the athlete is running at maximal or close to maximal speeds [6,16]. In athletic training with repeated sprint runs, sprinters experience muscle fatigue. If hamstring muscles become fatigued, the strength and flexibility will be reduced [16]. Furthermore, muscle fatigue may lead to the dys-synergic contraction of the different muscle groups [10]. Therefore, sprinters generally suffer from hamstring injuries when practicing. It has been a serious problem for sprinters and their coaches for numerous years.

To the best of our knowledge, no studies have focused on the influence of preventive programs that take muscle fatigue into consideration in sprint events. We investigated the relationship between the number of hamstring injuries that occurred during submaximal–maximal/supramaximal running and the content of the prevention programs implemented. Submaximal and/or maximal running for usual (non-assisted) training results in a large number of runs leading to muscle-fatigue-related consequences. Furthermore, supramaximal running for special (assisted) training [17,18] involves a small number of runs with low-muscle-fatigue conditions.

The majority of modeling studies do not test for the influence of fatigue, which is likely to be a major hamstring injury risk factor [19]. In this study, we examined the number of hamstring injuries caused by different numbers of runs. This study aimed to clarify how the muscle fatigue condition affects the success of prevention programs in sprinters.

## 2. Materials and Methods

### 2.1. Subjects

The initial subject cohort consisted of 649 sprinters recruited from a track and field sprint team; 36 sprinters were excluded from the study because of various reasons, such as transfer to other events, becoming a physical trainer or assistant manager, and withdrawal from the team. In this study, the enrolled subjects comprised 613 collegiate male sprinters (aged 18–24 years) over the course of 24 track and field seasons. Participants who had a medical history within six months of study were carefully monitored. Some of them were top-ranked sprinters at intercollegiate or national Japanese championships. The participants included a member of the Japanese 4 × 100 m relay team (fourth place overall) from the XXVIII Olympic Games (2004, Athens) and a member of the Japanese 4 × 100 m relay team (second place overall) at the 23rd Universiade Games (2005, Izmir).

All subjects were voluntary participants from the same track and field sprint team. The sprint training program was supervised by the same coach for all 24 seasons. The coach, a co-author of this paper, formerly from Juntendo University, also had the final decision on sprinters’ participation in practices and at competitions. Complete records were maintained on practices and competitions for each sprinter. This study was approved by the institution. Informed consent was obtained from the subjects before their participation.

### 2.2. Study Design

Figure 1 presents the practical study design. The research took place over 24 track and field seasons. In 1988, tow training involving repeated episodes of supramaximal running was first introduced as a special program for our sprint team. Since then, three types of sprint trainings have been practiced: submaximal, maximal, and supramaximal running. We conducted these sprint trainings safely and effectively.

The 24 seasons were divided into three periods: Period I covered four seasons (1988–1991), Period II covered eight seasons (1992–1999), and Period III covered 12 seasons (2000–2011). New training methods and new machines were developed and introduced for the Olympic Games of Seoul, Barcelona, and Sydney; as a result, the number of programs increased and prevention programs evolved [6,7]. Data from 2013 have been excluded because the coach stepped down from the sprint team.

### 2.3. Prescription for Quality/Volume of Overall Sprint Training throughout the Year

Figure 2 presents the standard program with usual and special training arranged for the overall sprint training throughout the year. The training aimed to acquire the maximal running speed and speed endurance. To achieve the goal, the annual training program was divided into six phases. These phases were recovery, basic training, preseason, first season, inter-season, and second season. Standard sprint training for submaximal and maximal running was continuously practiced in five phases, excluding the recovery phase. In contrast, special training for supramaximal running was practiced during limited weeks in four phases, excluding recovery and basic phases. Three types of running, namely, submaximal, maximal, and supramaximal, were performed throughout the year, excluding the recovery phase, with adjustments in quality and volume. In terms of the individualized training of elite sprinters, they had trained with other sprinters and not individually. Moreover, one of the most crucial things in coaching was not to practice at maximal effort so that sprinters could remember to stay relaxed during any running speed training.

### 2.4. Submaximal and Maximal Running

The purpose of usual training throughout the year is to acquire maximal running speed and speed endurance. Usual training for submaximal and maximal running has a sufficient volume (normal training) to cause overload, which is followed by acute fatigue. Usual training helps sprinters achieve maximal running speed and speed endurance by repeated acute fatigue, which causes an adaptive response.

The objective of usual training for repeated submaximal and maximal running in each phase was to strengthen the comprehensive physical fitness, particularly legs and strength-endurance in basic training, to increase maximal running speed in preseason, keep maximal running speed in the first and second seasons, and regain maximal running speed in the inter-season (See Figure 2). Submaximal and maximal running were practiced with independent efforts (non-assisted) or under an increased workload, such as running uphill or using a sled (resisted). Therefore, the usual training for submaximal and maximal running comprises a large number of runs (Table 1).

### 2.5. Supramaximal Running

Supramaximal running is a type of quality training; therefore, it was conducted at the point in the overall sprint training schedule when quality was being improved or when quality was high. Moreover, the total volume of training decreased (Figure 2). Supramaximal running can be used as an effective sprint training method specific to sprinters to improve mechanical power for hip extensors and knee flexors in hamstrings [17]. The higher levels of a neuromuscular activity achieved by supramaximal running and the related higher stride frequency work as an auxiliary method to improve the level of maximal sprinting and prepare for a competition. Special sprint training for supramaximal running includes a small number of runs (Table 2). Muscle fatigue is a risk factor for hamstring injuries [10,19,20]; therefore, supramaximal running was performed following a day off or on an individual practice day when muscle fatigue was expected to be at its lowest. Each sprinter participated in 2–5 runs/day for 15–25 days/season. The number of runs per sprinter over a season was limited to a maximum of 50. In reality, no one reached 50 runs; their number was from 20 to 30. The supramaximal running was adjusted appropriately so that it could be practiced on individualized principles.

In special training, supramaximal running has been used as an artificial means to pull forward and make athletes sprint for a certain period of time when maintaining a supramaximal speed [6,7,21]. Supramaximal running was practiced with assistance in tow training. Tow training was performed using a towing machine (Figure 3) and a rubber tube [6,7,21]. The towing machine allowed sprinters to be towed for more than 100 m. The rubber tube allowed numerous sprinters to be towed for up to 50 m. Two types (50 and 100 m) of supramaximal runs followed by a 20 m approach run with a smooth starting method [22,23].

The percentage increase in the velocity of a sprinter during supramaximal running is up to 110% [18,21,24,25,26,27,28,29,30]. During our training, the rate was often set at 103–107% [17,21,28,29,30,31]. Within this range, the sprinter could acquire a high stride frequency in a subjective sprinting motion. Most importantly, the sprinters in this study were instructed to practice active landing by stamping the landing leg as rapidly as possible onto the ground [24]. We explained the purpose of tow training to the sprinters to allow them to practice (implement) supramaximal running with an awareness of higher stride frequency. To verify whether subjects followed this instruction, test runs for supramaximal running were performed [7].

### 2.6. Hamstring Injury Prevention Program

The injury prevention program followed by the sprinters has evolved over time to reflect the current most effective strategies for preventing hamstring injuries (Table 3). Period I comprised only concentric hamstring strengthening with a traditional leg curl weight machine. In Period II, agility training, such as ladder and mini-hurdle exercise, was added to the program of Period I. Moreover, in the middle of Period II, a newly developed weight machine was introduced, which enabled a concentric hip extension exercise. In Period III, in addition to the programs taken place in Period II, eccentric hamstring strengthening exercises (Nordic hamstring exercise [1,2], glute–ham raise exercises, [32] and dynamic stretching exercises) were added.

The sprinters practiced the program in compliance with the load, action, and motion designated for each program as mentioned in Table 3. The program used was adjusted in each case according to the judgment of the coach and with sufficient consideration for the condition of the given sprinter. Strength training was performed as a part of the weight training. Agility and flexibility training was performed during individual warm-ups.

To confirm whether the hamstring injury prevention program was effective [10], the coach and sprinters periodically investigated the effects of the prevention program. Muscle strengthening was objectively verified by increased ability (e.g., weight, repetitions, time, and the number of sets) in weight training. Neuromuscular function improvement was objectively verified by the decreased time required for each sprinter to clear 15 miniature hurdles at the lowest height established at fixed intervals. Dynamic flexibility (measure of resistance to active motion around a joint or series of joints) [33] was determined by whether the sprinting motion, involving hip, knee, and ankle joints, was performed smoothly and quickly. The determination was solely based on the subjective opinion of the coach and sprinters.

### 2.7. Definition of Hamstring Injury

Throughout the season of 24 years, the team trainer and physician, Japan Sports Association (JASA) sports doctor, documented all hamstring injuries resulting from a practice with submaximal, maximal, and supramaximal running. Hamstring injuries were diagnosed by local tenderness, pain, and reduced range of motion on the straight leg raise test, and by evaluating for pain and reduced strength during resisted knee flexion while prone [34]. We defined an incident of hamstring injury as one that caused the sprinter to refrain from training or competition for at least one week [35].

### 2.8. Statistical Analysis

According to a prevention strategy, we recorded the number of sprinters who experienced hamstring injuries in each of the three periods. Based on previous studies, the injury rate for each period was calculated as the number of hamstring injuries per athlete season. Injury incidences were calculated as the numbers of hamstring injuries per 1000 sprinters [36,37,38]. The incidences of hamstring injury in Periods Ⅰ, Ⅱ, and Ⅲ were examined using the chi-squared test. Chi-squared tests were performed using SPSS version 27 (IBM Corp. Armonk, NY, USA).

Effect sizes (ESs) and power in post hoc tests were calculated using Gpower3 (version 3.1) [39]. ES was used as (w). ES (w) strength was rated as small (0.10), moderate (0.30), and large (0.50) [40], and α error was set to *p* < 0.05, whereas β error was set to (1 − β) > 0.80.

### 2.9. Ethical Consideration

The human ethics committee of Juntendo University approved the protocol for this research. Informed consent was obtained from all subjects.

## 3. Results

Figure 4 shows the hamstring injury rate during submaximal, maximal, and supramaximal running over the study period. The incidence of hamstring injuries during submaximal and maximal running per season was 103.4 for Period I, 131.3 for Period II, and 183.9 for Period III, among which no significant change was found [X^2^(2) = 5.15, *p* = 0.07, ES(w) = 0.26, (1 − β) = 0.99)]. In contrast, the incidence of hamstring injuries during supramaximal running per season was 137.9 for Period I, 60.6 for Period II, and 6.7 for Period III, with the incidence decreasing significantly [X^2^(2) = 31.78, *p* < 0.01, ES(w) = 19.09, and (1 − β) = 1.00].

## 4. Discussion

To implement tow training with a small number of runs, the combination of prevention programs showed positive results. Conversely, in usual training with a large number of runs, the combination of prevention programs showed no positive results.

### 4.1. Consideration of the Methodology

It was important for the value of this study that one coach led the university track and field team (i.e., the subjects of the study) for 24 years, during which the team remained competitive at the national level. Therefore, the 613 sprinters investigated can be considered to be uniform. Moreover, the sprint training that employed sub/maximal running and supramaximal running was of excellent quality.

Sprinters should consider learning how muscle fatigue affects the success of preventive programs rather than soccer and football players. In ball games, hamstring injuries are more likely to be caused by sharp twists, cuts, and physical contact, whereas in sprinting, injuries often occur when running in symmetrical and linear movements [41]. For these reasons, sprinters are more likely to identify the cause of hamstring injury than ball game players.

### 4.2. Effects of the Combination of Prevention Program on the Number of Hamstring Injuries during Supramaximal Running

The risk of injury increases in fatigue conditions [19,20]. Fatigue conditions may lead to dys-synergic contraction of the different muscle groups, lack of muscle strength, and decreased muscle endurance, causing hamstring injury [10]. Therefore, supramaximal runs were practiced with a small number repetitions because of the consideration given to the fatigue conditions of the sprinters.

All sprinters strengthened their hamstrings with leg curls, hip extensions, and two types of modified Nordic hamstring exercises. The power demonstrated by the eccentric contraction of the knee flexors and the concentric contraction of the hip extensor in the late part of the swing phase are reported to increase during supramaximal running [21]. Supramaximal running, wherein a sprinter sprints at a supramaximal level by artificial means, forces the hamstring to perform activity at extremely intense levels [17,21]. In sprinters, these generated high forces are postulated to be related to the hamstring injuries during supramaximal running.

The practice of strength training applies a concentric and an eccentric load to the hamstring through the trunk position at the injury position. Eccentric strength training worked well against a load on the hamstring during supramaximal running. In this study, the number of hamstring injuries decreased during Periods II and III with added concentric to eccentric strength training compared with Period I. These training programs reduced the number of hamstring injury cases during supramaximal running.

Positive neuromuscular changes can occur in special training for supramaximal runs, including increased muscle stiffness and efficiency of muscle contraction [18,30]. Furthermore, supramaximal running is an effective means of sprint training to improve the mechanical power for hip extensors and knee flexors in hamstrings [17]. Therefore, the leg movement becomes rapid in supramaximal running.

It is possible to achieve a higher stride rate during supramaximal running than with maximal running. However, the excessive stride rate increases the likelihood of hamstring injury [6]. Mero and Komi [24,27,30] highlighted that adequate neuromuscular performance is crucial for supramaximal running. Some neural networks are specifically required for high locomotor speeds [42]. Thus, an appropriate neural control is a key factor in preventing hamstring injuries during supramaximal running for a higher stride frequency.

Agility training is practiced as a means of learning the rapid movements required to cope with a supramaximal level of running. Sprinters trained on ladders and mini hurdles exhibited rapid strides at a speed equal to or faster than the observed strides (4.00–4.76 steps/sec) during supramaximal running [6]. Motion training at a high level, such as supramaximal running, which requires strong foot contact, requires the learning of new muscle recruitment patterns that involve a peripheral sensory input [7]. Using ladders and mini hurdles, sprinters may be able to obtain muscle synergies adapted to high levels of sprints. Therefore, the number of hamstring injuries during the supramaximal run was reduced.

Dynamic stretching should be a part of warm-up routines because of its similarity to the movement patterns of the subsequent activity in addition to the promotion of greater muscle activation [43] in supramaximal running. Performing dynamic stretching, the neuromuscular system acts as a softer musculotendinous system with an increased length and the ability to perform larger movements [44].

The objective of training with dynamic stretching is to acquire flexibility in the lumbopelvic region muscles and adapt the hip joint to a mobile state in supramaximal running, where dynamic flexibility is secured. The hip joint movement adapted to supramaximal running is likely acquired by stretching the hamstring, quadriceps femoris, and other muscles while actively moving the joints.

Dynamic stretching is conjectured to function effectively in preventing hamstring injury when assisting athletes to perform at a high level. Dynamic stretching exercises are potentially an effective way to prevent hamstring injuries in supramaximal running.

The outline of the 24-year tow training program has not changed. The combination of prevention programs, agility, strength, and flexibility training reduced the incidences of hamstring injuries. The prevention program was effective in the supramaximal run, which was practiced with a small number of runs with a consideration of the fatigue condition of the sprinter.

### 4.3. Effects of the Combination of Prevention Program on the Number of Hamstring Injuries during Submaximal and Maximal Running

Usual training has sufficient volume and intensity to cause overload; it is followed by acute fatigue. The summation of several training programs throughout the year will result in greater fatigue and a subsequent supercompensation response [45]. This adaptive response will take place, which will leave the subject in a healthier state than the previous condition [45]. Therefore, contents of a usual sprint training for submaximal and maximal running comprise a large number of runs. In this study, even if the program could a prevent hamstring injury occurring during sub/maximal running with a sufficient volume for a large number of runs in a usual training, the incidences of hamstring injury did not decrease.

Fatigue influences muscle activation and function, such as lumbopelvic control, knee stability, leg stiffness, and muscle–tendon unit energy transfer [19]. Alterations in running kinematics, such as a “Groucho position” due to fatigue, reduced exercise efficiency and increased force moments, associated with increased stress on contractile muscle units; theoretically, there is a risk of damage to hamstring muscles [19].

There is research on the relationship between muscle strength, agility, flexibility, and fatigue. In muscle strength, Small et al. [46] note the eccentric peak hamstring torque and the functional hamstring:quadriceps (H:Q) ratio (eccentric hamstrings versus concentric quadriceps) were significantly reduced during the fatigue-inducing protocol. This characteristic is more remarkable at faster contraction speeds [47]. In flexibility, active straight leg raises decrease after hamstring fatigue exercise [48]. In agility, muscle fatigue may reduce neuromuscular coordination [49]. Furthermore, in practical research, far more often than not, more hamstring injuries occurred at the end of each half during a ball game [11,12], which may indicate that fatigue is involved in long exposure times [12].

In usual training, which causes muscle fatigue with a large number of runs, the physical performance of increased strength, agility, and flexibility with prevention programs may not have a positive effect on hamstring activation and function. Our hamstring injury prevention program had no effect on the fatigued hamstring and the muscle groups, hip extensor, and knee flexor.

### 4.4. Suggestions for Training

In running-based training, clarifying and considering the set time for distance are important so as to not only improve performance but also prevent injuries. In usual training for a large number of runs, it is difficult to clear the set time because of fatigue caused by repeated running. At the time, “Groucho running” patterns because of poor muscle activation and function may be observed [19]. A sprinter who cannot clear the time set by the coach will not have achieved the purpose of the training. Therefore, for running-based athletes, these should be used to initiate careful monitoring of running time and inform training load so that the athletes are objectively exposed to the appropriate volume of training [20]. Furthermore, it is also important to subjectively consider fatigue. Coaches check the athlete’s physical condition (degree of fatigue) before training using a numerical value (e.g., the visual analog scale) to quantify the athlete’s physical condition. It may be important for coaches to encourage athletes for stopping training to monitor the degree of fatigue objectively and subjectively and to avoid the risk of injury.

Furthermore, high-intensity aerobic exercises practiced in the basic training period should be considered within hamstring injury programs [50]. Retrospective evidence indicates that prior hamstring injury is associated with strength-endurance deficits [51]. The improvement of overall fitness levels reduced the burden of fatigue [20].

No matter how effective the training content if a risk of injury exists, it is unacceptable. Careful monitoring is required to ensure that the sprinter is exposed to the appropriate volume of training that is effective in avoiding spikes in training load and preventing the development of hamstring injuries [20].

### 4.5. Limitation and Future Direction

When a sprinter is sprinting at maximal or close to maximal speeds, hamstring injuries often occur when running in symmetrical and linear movements. However, in ball games, hamstring injuries are more likely to be caused by sharp twists, cuts, and physical contact. Therefore, it remains questionable whether the suggestions of this study are applicable to ball game players.

This study suggests that muscle fatigue affected the hamstring injury prevention program. However, because the method for measuring muscle fatigue has not been established, the degree of fatigue cannot be confirmed. In the future, it is necessary to develop a method that can objectively measure the factor of muscle fatigue-related consequences, and investigate whether muscle fatigue affects the onset of hamstring injury.

## 5. Conclusions

For a small number of runs for sprint training, the hamstring injury prevention program functions well. Muscle fatigue conditions with a large number of runs improve the effectiveness of prevention programs for those vulnerable to hamstring injury.

## Figures and Tables

**Figure 1 ijerph-19-09375-f001:**
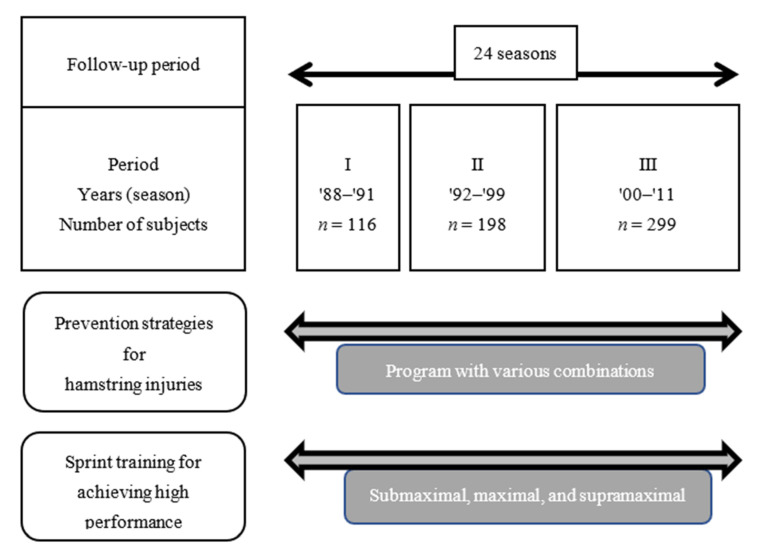
Study design.

**Figure 2 ijerph-19-09375-f002:**
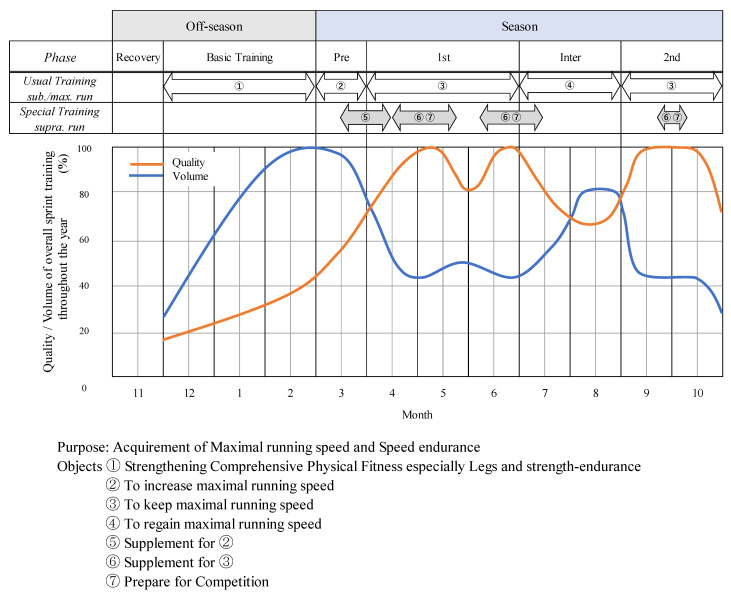
Concept of overall sprint training program throughout the year.

**Figure 3 ijerph-19-09375-f003:**
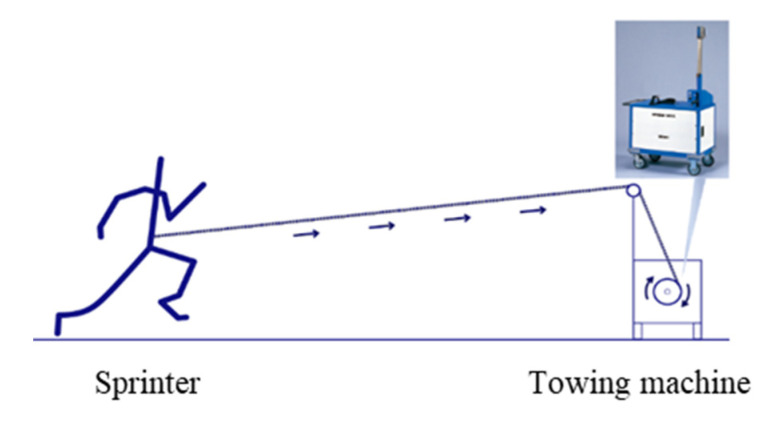
Supramaximal running in towing system. The towing machine comprises an electric motor, a powder clutch, and a dram to wind rope. A powder clutch works to stabilize the tension to pull the runner safely.

**Figure 4 ijerph-19-09375-f004:**
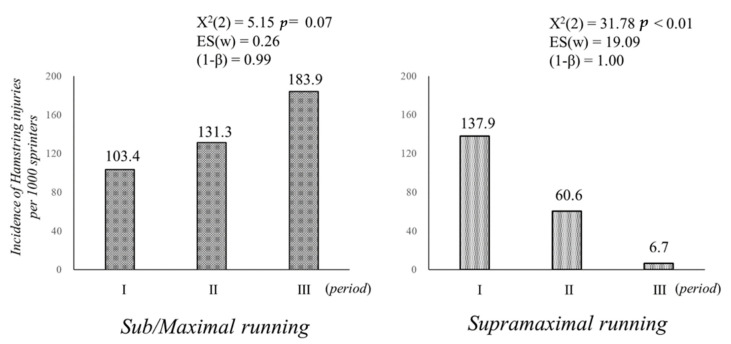
Hamstring injury rate during submaximal, maximal, and supramaximal running.

**Table 1 ijerph-19-09375-t001:** Contents of usual sprint training for submaximal and maximal running in a large number of runs.

**Basic Training: ①**	**Preseason** **: ②**
30 min build up every 10 min for Cross-Country 100 m at 70% OR 200 m at 60% × 10–20 repetition for Up Hill Running 100–150 m 10-kg-weights × 5 repetitions on Sled for Resistance Running. 200–300 m at 60% × 10 repetitions 250–400 m at 60% × 10 repetitions	50–100 m 10-kg Weights × 5 repetitions on Sled for Resistance Running 150–200 m at 80%–90% × 3 repetition × 2–4 sets 300 m at 70%–80% × 5 repetitions 1–2 sets
**1st and 2nd Season: ** **③**	**Interseason: ** **④**
50–100 m 5-kg Weights × 2 repetitions on Sled for Resistance Running 30–60 m at 90%–100% × 5 repetitions for Start Dash and (100 m at 95%, 150 m at 95%, 200 m at 90%) × 1–2 sets 30–60 m at 90%–100% × 5 repetitions for Start Dash and (100–120 m at 95%) × 3–5 repetitions, 50 m at a constant tempo for Skip × 5 repetitions 30–60 m at 90%–100% × 5 repetitions for Start Dash and (200 m at 90%, 400 m at 90%) × 1–2 sets 30–60 m at 90%–100% × 5 repetitions for Start Dash and (250–300 m at 90%) × 3–5 repetitions, 50–100 m at a constant tempo for Skip × 3–5 repetitions	50–100 m 5-kg Weights × 2 repetitions on Sled for Resistance Running 150 m at 90% × 3 repetitions × 2–3 sets (150–200 m at 85%) × 5 repetitions, 50 m at a constant tempo for Skip × 5 repetitions 250–300 m at 90% × 3 repetitions × 1–2 set (200–300 m at 85%) × 3–5 repetitions, 50–100 m at a constant tempo for Skip × 3–5 repetitions

The sprinters practiced as a sprint training of either content during each phase; Percentage represents the rate of increase in running velocity.

**Table 2 ijerph-19-09375-t002:** Contents of Special sprint training for Supramaximal running in a small number of runs.

**Pre-Season and 1st Seasin: ** **⑤**	**1st, 2nd and Inter-Season: ** **⑥** **, ** **⑦**
50–100 m at 105% × 3–5 repetition	50 m at 105–110% × 3 repetition 50 m at 105–110% × 1

The sprinters selected the distance, the rate of increase in running velocity and numbers in towing during each phase.

**Table 3 ijerph-19-09375-t003:** Description of a preventive standard program for hamstring injury.

Objective and Method	Action and/or Motion (Load)	Period		
		I	II	III
Strength				
Weight macine	Knee flexors concentrically (leg curl)	●	●	●
	(3/5-4/5 of body weight × 10 repetitions × 3–5 sets)			
	Hip extensors concentrically (hip extension)		●	●
	(4/5-5/5 of body weight × 10 repetitions × 3–5 sets)			
Body weight	Knee flexors eccentrically (Nordic hamstring exercise)			●
	(lean forward slowly × 30–60 seconds × 5 sets)			
	Knee flexors eccentrically and hip extensors/knee flexors concentrically (glute-ham raise)			●
	(lean forward, downword, and upward × 10-20 repetitions × 5 sets)			
Agility				
Ladder	5 types of fast stepping in all directions		●	●
	(10 m × 4 repetitions)			
Mini-hurdle	4 types of one and/or both leg(s) with fast stepping		●	●
	(10 hurdles × 4 repetitions)			
Flexibility				
Dynamic stretching	3 types of stretching for muscles around hip joint			●
	(20 m × 1 repetitiion)			

## Data Availability

The data presented in this study are available on request from the corresponding author. The data are not publicly available due to individual privacy reason.

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
