# Peer review of "Effects of Various Numbers of Runs on the Success of Hamstring Injury Prevention Program in Sprinters"

_ijerph, 2022, doi:10.3390/ijerph19159375_

Round 1
Reviewer 1 Report
Overall comments:
I commend the authors for embarking on such a large project—this is undoubtedly a lifetime of work for some of the authors, and a wealth of information is provided. The following comments are intended to improve the quality of the study to ensure that other coaches, clinicians, and researchers can adequately benefit from your years of hard work.
The greatest concern regarding the study is that it is unclear what variables are being manipulated between periods. It’s also unclear what the authors believe made the difference. The paper is written in a way that seems to indicate that the supramaximal running, if timed correctly, is helpful. However, a hamstring injury prevention program was also implemented during these time frames, and this program appears to have evolved over time. Additionally, a lot of the discussion is spent on the effects of fatigue—which leaves readers even more confused about the purpose of the paper. Though the hamstring injury incidence decreased in the last period, it’s unclear what changed between periods. With so many variables changing over time, how do we know what caused the change in hamstring incidence? Additional detail regarding period distinction would be helpful (ex: why was period 3 triple the length of period 1?).
The analysis is too simplistic for the number of variables discussed in the paper (see next paragraph for suggestions); for a longitudinal design like this, at the very least time needs to be considered as a variable.
There are so many variables discussed in the paper—tow training, injury prevention program, supramaximal running, fatigue state—it’s unclear which is meant to be an outcome and which is an intervention. My suggestion to the authors is to break this paper up into multiple papers—this would allow more time/space to describe in greater detail what was done during training periods (paper 1), however the injury prevention program evolved over time (paper 2), and how appropriate fatigue can improve performance and decrease injury (paper 3). This would also allow for multiple, more sophisticated analyses to create more definitive and appropriate conclusions.
There are several strengths to this study, however, there are too many concepts to be presented in a single paper.
72: Were all participants included for the same amount of time? Did anyone drop out due to illness or injury? Or because of transfer to another or dropping out of university? (this happens commonly in the United States, though it may not be as common in Japan)
77: What does low running skill mean? Could you provide some additional detail here?
92: Was this previous study published? If so, please cite. If not, please clarify. Additionally, please clarify the findings—was it truly safe and effective?
93-95: Why were the seasons broken up into periods in this way? I understand combining seasons for easier comparison, but what was the rationale for blocking them in this way?
124: Space between Figure and 2 (not Figure2); additionally—do the concepts presented in figure 2 represent what was done during the entirety of that last 24 seasons? I would imagine that with the advances in wearable technology, some of the training concepts have evolved in the last quarter of a century. Additionally, I would suspect that with elite runners (like the Olympians described), much of this training is individualized. Are the contents of Figure 2 just an example of what training looked like for a specific runner? And during what Period (I, II, or III)?
138: Space between Figure and 2 (not Figure2)
139-140: Edit for clarity—I believe this may be a translation issue, but I’m not sure what this sentence is saying
144: missing a space after the period and before Since
150: Here you talk about the individualization of the process—I think this would be an important concept that should be introduced a little earlier
175-176: Suggested edit: “The injury prevention program followed by the sprinters has evolved over time to reflect the current most effective strategies for preventing hamstring injuries (Table 2).”
189-190: If we are determining the effect of an injury prevention program, wouldn’t the best metric be to count the number of injuries? While strength, neuromuscular function, and flexibility have been cited as risk factors for hamstring injury, the purpose of the program is to avoid injury. Additionally—do you have references to support these outcomes as valid? This is of particular concern for the neuromuscular function and dynamic flexibility.
208: How were injuries documented? It appears that injuries were categorized based on what type of running the athlete was performing at the time of injury—who documented this? How were these records kept over the last 24 years?
225: Given the number of variables that are present, this appears to be too simplistic of an analysis. Most notably, time is not included as a variable in your analysis. How do we know that injury rates won’t change just by chance? Additionally, there is no control for potential confounders, such as the evolution of the hamstring injury prevention program over time.
262-263: Improve hip extensors and knee flexors…what? Strength? Neuromuscular control? What sort of improvement is seen in these muscle groups?
290-291: The changes in training between periods is not clear.
292-294: Again, if all sprinters did this, how do you know this decreased injury rates in the last period? What evidence suggests this relationship?
305-312: Though your injuries during supramaximal running appear to have decreased, the link between this outcome and the injury prevention program is not clear. Additionally, though not statistically significant, you did observe a steady increase in injuries during submax and max running—why is this not discussed?
345: change to “because of fatigue”
365: These conclusions cannot be supported by the evidence you provided.
Reviewer 2 Report
General comments
Overall, I feel that while the topic of this research is important, and that understanding the effect of fatigue on sprint performance and injury rate is worthy, the present study does not explore these concepts with enough clarity. Many claims are made, yet there appear to be very few tests of performance and some things such as "fatigue" are not quantified or measured.
Specific comments
There is a need to state how fatigue was measured and how it was kept constant relative to the individual while training. It is not certain if participants experienced similar "fatigue".
The line "Dynamic flexibility (measure of resistance to active motion around a joint or series of joints) [37] was determined by whether the sprinting motion was performed smoothly based on the subjective opinion of the coach and sprinters themselves" needs further explanation. This does not seem scientifically rigorous.
How were historic injuries recorded? At the time? Was missed training recorded when the injury originally occurred or were participants asked to recall it?
Section 4.2 concerns me, as there are a lot of inferences made, but none tested. It has been claimed that neuromuscular control, agility, power and flexibility improvements have all added to the resilience of the hamstrings, but that has not been proven, just speculated.
in sections 4.3 and 4.4 it is posited that athletes when fatigued display the "Groucho position" when running, but was this tested at any point? Did any athlete display this? Were there kinematics observed?
Reviewer 3 Report
The manuscript fits well to the scope of the journal submitted to. The objective of this study was to clarify how muscle fatigue condition affects the success of prevention programs in sprinters..
The paper is well structured and organized. The introduction is clearly written and references to the literature are correct. The methods are generally well-conducted and reported.
The results obtained in this study will be helpful to coaches and athletes in their routine work.
additional comments
The main question addressed by the research is how muscle fatigue condition affects the success of prevention programs in sprinters.
In my opinion, the problem of preventing injuries in sport is very important. In competitive sport, more and more are required of players. The development and validation of injury prevention programs is an important component of sport research. The program proposed by the researchers covers observations since 1988. It is an extensive material based on the research of leading players, whose training loads are significant. Therefore, I believe that the contribution of the presented research to the subject of muscle fatigue and the impact of training is significant.
Conclusions are consistent with the evidence and arguments proposed and they address the main question posed.
The refferences are appropriate.
The tables are legible and facilitate the reception of the text. The figures well illustrate the information contained in the text.
Author Response
Dear reviewer 3,
Thank you for your time revising my manuscript. As you mentioned, I believe this research will contribute progress the method of prevention of hamstring injury.